# Crack Width Estimation of Mortar Specimen Using Gas Diffusion Experiment

**DOI:** 10.3390/ma12183003

**Published:** 2019-09-16

**Authors:** Do-Keun Lee, Min-Hyuk Lim, Kyung-Joon Shin, Kwang-Myong Lee

**Affiliations:** 1Civil Engineering, Chungnam National University, Daejeon 34134, Korea; likedg@naver.com (D.-K.L.); limmh0701@naver.com (M.-H.L.); 2Civil, Architectural and Environmental System Engineering, Sungkyunkwan University, Suwon 16419, Korea; leekm79@skku.edu

**Keywords:** gas diffusion coefficient, cracked specimen, crack width estimation

## Abstract

Maintenance of structures using self-healing concrete technologies has recently been actively studied. However, unlike the technological development of self-healing concrete, research focused on evaluating the self-healing performance is insufficient. Although water permeability experiments are widely used, the reliability of the test results may be reduced due to the viscosity of water and the possibility of elution of material inside the specimen. In this study, we propose a gas diffusion test for estimating the crack width and eventually for application to evaluation of the self-healing performance. The results verified that the proposed method can be effectively applied to the estimation of crack width.

## 1. Introduction

Cement-based materials such as concrete, mortar, and cement paste have been used for a long time as major construction material. However, these materials are characterized by a relatively low tensile strength and are thus prone to cracking due to several reasons, such as volume change, evaporation of water, and excessive stress of concrete [1]. For concrete structures, it is almost impossible to completely restrict the crack formation using convectional reinforcements due to the nature of reinforced concrete structures. Cracks endanger the durability of the structures as aggressive moisture and chemicals can infiltrate the matrix via the cracks, and they thus cause deterioration of the usability of the structure and increase the maintenance cost [2,3].

Meanwhile, as the technology in the field of concrete further develops, various materials capable of repairing and reinforcing the cracks in concrete structures are being developed and used to minimize the deterioration by cracking. Numerous studies have recently been carried out on the development of “self-healing concrete”, which restores cracks using existing materials inside the concrete without the need for further external artificial manipulation [4,5,6,7,8,9]. 

In order to evaluate the self-healing performance of self-healing concrete, it is necessary to prepare specimens with a specified crack width and let them cure in order to confirm whether the crack is properly repaired. Since the crack width itself is the basis of the performance evaluation, this procedure requires an accurate and reasonable measurement of the crack width in the specimen.

Although the crack width in the specimen can be primarily measured with an optical microscope, it is difficult to reflect the characteristics of irregular cracks by visual measurements. Moreover, the variation of the crack width inside during the healing process cannot be measured by only observing the surface crack width. Therefore, this optical method can be used only for limited applications because the observation is limited to cracks on the outer surface [10]. Researchers have also suggested measuring the cracks by acquiring the shape of the entire specimen with the use of X-rays and computerized tomography(CT) images. However, these approaches have limitations, such as difficulty to distinguish between cracks and pores and costly measuring equipment. Meanwhile, the permeability test is the main method for indirectly evaluating the filling of cracks in the specimen. The permeability test has the advantage of not requiring complicated expensive equipment. However, there are several drawbacks, such as the possibility of the elution of materials inside the crack, the inflow of foreign substances during the test, and unknown parameters influencing the test results, such as unexpected head losses and the viscosity of water flow [11,12]. 

Therefore, in order to overcome the drawbacks of the traditional permeability test method, this study developed a technique to estimate the crack width and healing performance of mortar and concrete by measuring the gas diffusion through the crack. In order to verify this technique, we first performed a gas diffusion test on acrylic specimens with idealized straight cracks and analyzed the correlation between the crack width and diffusion characteristics. Based on Fick’s Law, this study showed the relationship between the amount of oxygen diffused, the thickness, and the crack width of the specimen. Finally, a gas diffusion test was performed on mortar specimens with a crack in order to examine and verify the applicability of the gas diffusion test to actual mortar specimens.

## 2. Literature Review

### 2.1. Gas Permeability Experiments for Concrete Specimens

Gas permeability experiments have been used as an indirect measure to evaluate the durability of cementitious materials [13,14,15,16]. Because of the high penetration resistance of the cementitious material, the experiment proceeds very slowly under natural conditions, and therefore, experiments are usually carried out using high-pressure gas as a medium. Compared with experiments with a water, a gas diffusion test has a strong advantage in that that there is no physical clogging, expansion due to reaction with incoming water, or continuous hydration [17]. Many researchers have evaluated the transportational characteristics of cementitious materials using gas [16,18]. 

Recently, much effort has been invested in developing gas permeability and diffusivity experiments [16,19,20], and researchers have revealed that the size of the internal pores and the degree of saturation of the specimen during the gas penetration test are the major factors affecting gas permeability in noncracked specimens [21,22,23]. 

Similarly, gas permeability experiments on cracked concrete specimens have been carried out by applying pressure. The gas permeability was estimated by measuring the amount of gas passing through the cracks at the boundary of the specimen with various crack sizes [24,25,26]. Unlike noncracked specimens, the degree of saturation in the specimens did not have a significant effect on the gas permeability of cracked specimens, since most of the flow occurs through the crack [18]. 

### 2.2. Performance Evaluation of Gas Diffusion in Self-Healing Concrete 

As discussed in Section 2.1, gas diffusion in a cracked concrete specimen occurs primarily through the crack, so the effect of porosity is considered to be minimal. Thus, in order to evaluate the performance of self-healing concrete with a crack, specifically, to observe the degree of healing of cracks, a gas diffusion experiment can be carried out by applying the concentration gradient only without applying a pressure difference using complicated experimental devices as done in previous studies. 

Therefore, with an assumption that the gas diffuses only through the crack, we proposed a gas diffusion test method that is operated not only by a pressure gradient but also by a concentration gradient, especially for the performance evaluation of self-healing concrete. In addition, it should be noted that by not applying the pressure gradient to the crack surface of self-healed concrete, damage such as peeling of the filled material [27] can be prevented.

Another advantage of the gas diffusion experiment is that the medium has a very low viscosity compared with water. Energy loss occurs due to viscosity in a typical fluid flow [10]. In terms of the permeability experiment, as the crack width and flow rate are not linear, the higher the flow velocity is, the greater the effect on the head loss [25]. However, in the case of the gas diffusion test with oxygen, as the viscosity of oxygen (2.04 × 10^−5^ Pa s) is about 1/50 of that of water (1.02 × 10^−3^ Pa s) under general laboratory temperature conditions (20 °C), while the energy loss due to viscosity is very small and has little effect on the test results. 

### 2.3. Modeling of Diffusion Equation

#### 2.3.1. Ideal Gas State Equation

Ideal gas means a hypothetical gas in which the volume of the particles constituting the system is almost zero and there is little interaction between particles, and thus, the intermolecular potential energy is not important, and the intermolecular collision is a completely elastic collision [28]. The ideal gas law describes the relationship between temperature, pressure, and density, which represent the thermodynamic state of the atmosphere and can be expressed as Equation (1):(1)PV=nRT,
where P is the pressure (kPa), V is the volume (m^3^), n is the number of moles of the gas, T is the absolute temperature (K), and R is the gas constant and has a value of 8.3143 m^3^∙Pa∙K^−1^∙mol^−1^.

Meanwhile, considering that the ideal gas state equation is a law on the thermodynamic state of a single gas and that the total volume of gas has a linear relationship with the concentration, the total volume of gas can be defined by Equation (2) below:(2)$$V=\frac{\mathrm{nRT}}{P}\cdot C,$$
where C is the concentration. According to the boundary conditions of the tests performed in this study, C can be expressed as the concentration gradient between the inside and outside of the vessel. 

The following Equation (3) is obtained by differentiating Equation (2) with respect to time.

(3)$$\frac{\partial V}{\partial t}=\frac{\mathrm{nRT}}{P}\cdot \frac{\partial C}{\partial t}$$

#### 2.3.2. Fick’s Law of Diffusion

Fick’s law states that the amount of gas diffusion that occurs during a unit time through a unit area is perpendicular to the direction of diffusion [28,29]. Since the diffusive flux is proportional to the gradient of concentration, Fick’s law can be expressed by Equation (4):(4)$$\frac{\partial V}{\partial t}=-K\cdot A\cdot \frac{C}{d},$$
where K is the diffusion coefficient (mm^2^/s) and d is specimen thickness. 

The combination of Equations (3) and (4) is expressed as Equation (5) after a series of calculations [30]:(5)$$K=\frac{\mathrm{Vd}}{A_{c}t}\cdot \ln\left(\frac{C_{0}}{C}\right),$$
where $C_{0}$ is gas concentration at t = 0. 

When the area of specimen (A_s_) is used instead of the crack area (A_c_) [31,32], K can be regarded as the permeability coefficient per unit area of specimen (K_s_). Alternatively, when a length of crack (L_c_) is substituted for the crack area [27], K can be regarded as the permeability coefficient per unit length of crack (K_c_), and the volume (V) can be expressed as V_v_ by substituting the volume of the vessel, as shown in Equation (6). Since K_c_ and K_s_ are directly proportional to the crack width, they can be used for estimating the crack width.

(6)$$K_{c}=\frac{V_{V}d}{L_{c}t}\cdot \ln\left(\frac{C_{0}}{C}\right)$$

For the purpose of estimation of the crack width, Equation (6) can be rewritten as Equation (7) using the permeability coefficient per unit length of crack (K_c_). In this study, K_c_ is defined as a crack diffusion coefficient for convenience:(7)$$w={\alpha \;K}_{c},$$
where w is crack width and α is a proportional coefficient relating w and K_c_.

## 3. Experimental Work

### 3.1. Test Plan and Parameters

The test procedure consisted of two steps. The initial step was to identify the correlation between the width of an ideal crack and the gas diffusion coefficient. Twenty-five acrylic specimens with an ideal crack were made for this test. The crack width of the specimens ranged from 0.1 to 0.5 mm, and the thicknesses of specimens were from 10 mm to 50 mm. Table 1 shows the specimen IDs and the crack width of each specimen measured by microscope observation.

The second step was to explore the correlation between the width of a real crack in mortar and the gas diffusion coefficient. Thirteen mortar specimens possessing crack widths from 100 ± 10 to 380 ± 10 µm were prepared. The thickness of the specimens was 20 mm. 

### 3.2. Preparation of Test Specimens

In order to prepare a specimen with an ideal crack, acrylic specimens were made instead of using concrete or mortar, as shown in Figure 1. The specimens were prepared by attaching two acrylic blocks and inserting two thickness gauges at the edge of each side of the blocks. The thicknesses of the inserted gauges were 0.1, 0.2, 0.3, 0.4, and 0.5 mm. Since there were variations of the crack width due to unexpected errors resulting from the fabrication process, the average crack width of the specimens was measured at three locations on the upper and bottom surfaces using an optical microscope. The thicknesses of the specimens were 10, 20, 30, 40, and 50 mm.

Mortar specimens with a natural crack were made using a cylindrical specimen of Φ100 mm × 20 mm. The cylinders were split into two semicircular sections, as shown in Figure 2, using a crack inducing bar. A flexible silicone rubber sheet was then attached to both ends of the cracked sections in order to induce a crack of specified width. The desired crack widths were achieved by inserting silicone rubber sheets with varying thicknesses (0.1–0.4 mm). Finally, the split specimens were bound together using stainless steel bands to maintain the desired crack widths. Figure 3 shows the mortar specimens attached on the sealed vessels. Epoxy was applied to both sides of the specimen so that only oxygen could pass through the cracks. 

After the specimens were prepared, the widths of the cracks were measured using a microscope. Crack widths were measured at a total of six locations on the top and bottom surfaces, and then the averaged value was adopted as the crack width of the specimens. 

### 3.3. Experimental Apparatus and Method

A simple test apparatus was proposed for measuring the gas diffusion coefficient of the specimen. In order to create a gradient of gas concentration through the crack, one side of the specimen was exposed to air, and an airtight vessel was attached to the other side. By changing the gas concentration in the vessel, gas diffusion occurs due to the difference in concentration between two sides of the specimens. The internal dimensions of the sealed vessel were 125 mm × 125 mm × 60 mm and the surface attached to the specimen was cut off so that diffusion of the gas occurs through the crack and the hole of the sealed vessel, as illustrated in Figure 1. 

If the mortar specimen is saturated, moisture may be absorbed into the crack surface and may affect the measurement results. Therefore, as the experiment was performed with a dry crack surface, this study left the specimens under general laboratory conditions for 24 h to stabilize them. 

As a testing gas medium, this study adopted an oxygen gas, since it does not react with cement hydrate and is not toxic and easy to measure the concentration. These oxygen concentrations of the outside and inside of the vessel were monitored using an optical-based sensor, which can accurately measure the history of oxygen concentration without consuming oxygen or reducing the lifespan, with a resolution of 0.01%. For easy measurement, a wireless sensing unit based on Bluetooth communication was set up. 

The experiment was performed as follows. First, nitrogen was injected into the sealed vessel attached to the specimen to make the oxygen concentration inside the vessel close to 0%, which leads to diffusion of oxygen gas due to the gradient in the oxygen concentration inside and outside the vessel. The concentrations inside and outside the vessel were measured using oxygen sensors. This test was performed in a constant temperature and humidity room, which maintained a temperature of 20 ± 1 °C and a relative humidity of 60 ± 5%. Measurements were taken for about seven hours after injecting the nitrogen. It needs to be noted that the diffusion rate of the gas is closely related to the temperature so that the temperature-based correction would be necessary when the experiment is performed under the various temperature conditions.

## 4. Test Results and Analysis

### 4.1. Evaluation of Oxygen Diffusion Characteristics Using Acrylic Specimens

#### 4.1.1. Influence of Crack Width on Gas Diffusion

Figure 4a is a graph showing the oxygen concentration in the sealed vessel relative to the measurement time of C series specimens. As expected, the oxygen concentration in the sealed vessel increases and converges to the oxygen concentration in the atmosphere over time. However, the tendency of the value to change varied in relation to the crack width. As the crack width increases, the rate of increment increases, too. These results show that the amount of oxygen passing through the crack is closely related to the crack width, which corresponds to the size of the passageway.

In Figure 4a, the overall relationship between the concentration and time is not linear, since a condition of gas diffusion is an unsteady state. The concentrations inside and outside the specimens vary as time passes, and thus, the corresponding concentration gradient varies as well. However, as the measurement period becomes shorter, the increment of oxygen concentration becomes almost linear, as shown in Figure 4b. When the measurement time was 30 min, the coefficient of determination (R^2^) calculated for a linear function was larger than 0.99. 

When the relationship between concentration and time is linear, the diffusion rate can be regarded as being independent of time, which means that the gas diffusion rate is a constant. In this case, the gas flow can be expressed as a steady state, and the diffusion coefficient can be calculated directly from Fick’s 1st law. 

In order to verify the hermeticity of the test apparatus and a sample, the gas diffusion test was performed with an acrylic specimen without a crack. The result proves that there was only 0.005% of an increase in the oxygen concentration for 30 min.

#### 4.1.2. Influence of Specimen Thickness on Gas Diffusion

Figure 5 shows the variation of oxygen concentration in the vessel of the specimen with the same crack width (0.2 mm) but with a different specimen thickness (10–50 mm). The oxygen concentration in the sealed vessel increases up to the oxygen concentration in the atmosphere over time. It was observed that this convergence rate was affected by the thickness of the specimen. As the thickness of the specimen became thinner, the oxygen gas converged faster. 

Experiments on thickness also showed a linear behavior between the gas concentration and time when the measurement time was shortened. When the measurement time was 30 min, the coefficient of determination (R^2^) calculated for a linear function was larger than 0.99. 

#### 4.1.3. Oxygen Gas Diffusion Coefficient and Crack Diffusion Coefficient

With an assumption of steady state diffusion, the diffusion coefficient of oxygen gas through a crack can be calculated using Equation (6). According to Equation (6), the diffusion coefficient K is determined in relation to ln(C_0_/C) and time since the other variables are constant for a given specimen. Figure 6 shows the measured results based on ln(C_0_/C) and time and verifies the linearity of these parameters. Thus, it can be concluded that the diffusion coefficient can be calculated using the slope in Figure 6. 

Table 2 shows the calculated diffusion coefficient for the ideal cracks. It should be noted that the crack area is substituted into A in Equation (5) and the gradient of concentration means the difference in the concentration between inside and outside of the vessel. The calculated coefficients are in the range of 0.163 cm^2^/s to 0.227 cm^2^/s. The average diffusion coefficient of all of the tested acryl specimens was calculated as 0.199 cm^2^/s. The gas diffusion coefficient is a constant that typically is determined for the type of gas regardless of the test specimens. The results also show that the average measurement is within 2% of the generally known oxygen diffusion coefficient in air at 20 °C (0.203 cm^2^/s) [33].

Since the diffusion coefficient of oxygen gas is a constant, the crack width can be consistently estimated from the gas diffusion test when the crack length is known. Therefore, as explained in Section 2.3.2, the crack diffusion coefficient for oxygen gas is proposed as given in Equation (6). Table 3 and Figure 7 show the crack diffusion coefficients for the specimens of each series. Although there is a small amount of variation for a crack width of 0.5 mm, the crack diffusion coefficients were found to be directly proportional to the crack widths in all of the sections and ranges. This result can be interpreted as that the velocity profile of gas in the crack area is not parabolic but uniform and shows that the crack width can be estimated using the crack diffusion coefficient. 

#### 4.1.4. Estimation of Crack Width Using Crack Diffusion Coefficient

As shown in the previous section, it has been proven that the crack diffusion coefficient and the crack width have a strong linear relationship. Therefore, the possibility of crack estimation using a gas diffusion test is verified. 

The crack widths estimated using Equation (6) are listed in Table 3. The relative errors between the measurements and estimations were up to 19.34% and on average 6.81%. However, since the absolute error is within 0.08 mm (average 0.018 mm), it can be concluded that the predictions based on the crack diffusion coefficient are well fitted. 

### 4.2. Estimating the Crack Widths of Mortar Specimens Using Oxygen Diffusion Test

Similar to the results shown for the acrylic specimens, it can be expected that the crack width of the mortar specimens can be predicted well using a gas diffusion test. Since gas has very low viscosity, the roughness of the surface is supposed not to influence the results. The crack diffusion coefficients for cracked mortar specimens were calculated according to the same methodology applied for acryl specimens, as presented in Section 4.1.

Figure 8 shows the relationship between the crack diffusion coefficients and the crack widths of the tested specimens. The linear relationship between crack width of mortar specimens and the crack diffusion coefficient are observed. In order to compare the results of mortar specimens with those with acryl specimens, regression analysis was performed. The coefficient of determination (R^2^) calculated for a linear function is larger than 0.99. This implies that the linearity between the crack widths and the crack diffusion coefficient are valid for mortar specimens despite the presence of the roughness and irregularity of the crack surface. 

For the mortar specimens, the slope of the trend line (20.642) increased by 6% compared to that of the acrylic specimen (19.453). These results indicate that the diffusion coefficient in actual cracks is overestimated compared to the ideal linear cracks. There are two reasons under the premise that the gas velocity is not affected by materials such as saturation and porosity. First of all, the crack surface may be underestimated more than the actual crack length and crack width due to the unevenness of the crack surface. Second, when selecting the crack width, the average of the crack widths measured at 12 points per specimen was applied. Therefore, there may be a difference between the representative crack width and the actual crack width. Therefore, if we select representative cracks reasonably and narrow the difference with the actual crack width, the slope of the crack diffusion coefficient is not expected to be significantly different from that of the acrylic and mortar specimens. Therefore, to more accurately investigate the diffusion coefficient in actual cracks, the reasonable method for selecting crack width and the factors considering inhomogeneity should be examined in greater detail through further study. Nonetheless, it should be noted that this difference (6%) is much lower than that (about 50%) observed in a water permeability test [34]. 

Table 4 summarizes the crack widths measured using an optical microscope and the crack widths estimated using the gas diffusion test. The difference between the measurements and predictions was 0.007 mm or 4.12% on average. The maximum absolute error was 0.014 mm. This means that the crack width can be estimated within an error of 0.014 mm for these tested specimens.

## 5. Conclusions

This study proposed a gas diffusion experiment suitable for a cracked concrete. Using the proposed method, the relationship between the crack width and the oxygen gas diffusion coefficient was analyzed. Finally, a method to estimate the crack width of specimens was proposed. The following conclusions were obtained through this study. 

(1) A test method was proposed for gas to be diffused through a crack based on the concentration difference. Using this method, oxygen diffusion characteristics were investigated. As the time elapses after setting the initial concentration inside almost zero, the oxygen concentration inside the vessel increases and converges to the oxygen concentration in the atmosphere due to the concentration gradient between the inside and outside of the specimen. The overall relationship between the concentration gradient and time is not linear since a condition of gas diffusion is an unsteady state. However, when the duration is limited to within 30 min, oxygen diffusion through the crack can be regarded as steady state diffusion.

(2) The experimental results on acryl specimens with various crack widths show that the gas diffusion coefficient was similar regardless of the crack widths and specimen thicknesses. Even for the mortar specimens with a natural crack, the diffusion coefficients were similar. These results indicate that the crack surface does not influence the gas diffusion characteristic through the crack. 

(3) A crack diffusion coefficient is proposed for the estimation of crack width. The experimental results show that the crack diffusion coefficient of the specimen is highly related with the crack width. Using this relationship, the crack width can be estimated within an error of 0.08 and 0.014 mm for the acryl and mortar specimens tested in this study, respectively. This proves that the crack widths of specimens can be estimated well using an oxygen gas diffusive test. 

(4) Since the proposed method uses a gas as test medium, the influence of the viscosity of the test medium can be reduced compared to that of the water permeability test. In addition, the peeling and dissolution of material on a crack surface can be reduced. Thus, it can be effectively applied to estimate the crack width of self-healing concrete more than the water permeability test. 

## Figures and Tables

**Figure 1 materials-12-03003-f001:**
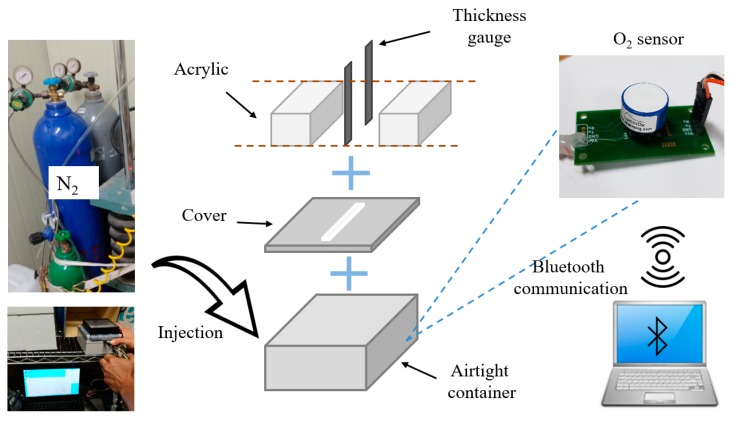
Outline of measurement device and method.

**Figure 2 materials-12-03003-f002:**
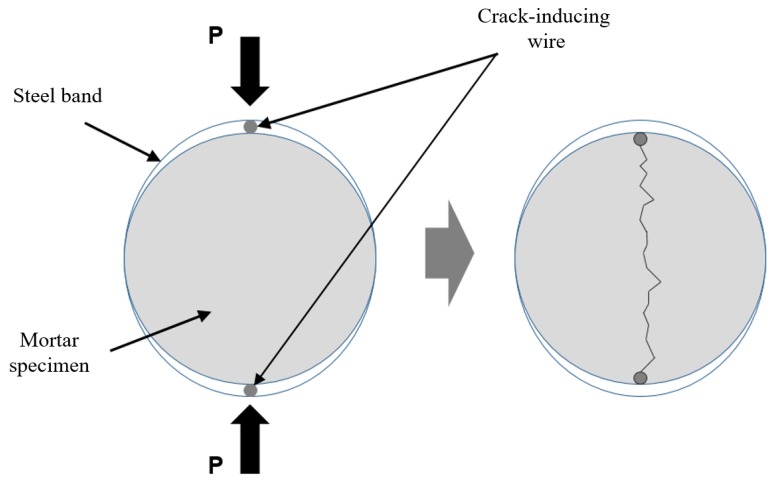
Scheme of splitting tensile test.

**Figure 3 materials-12-03003-f003:**
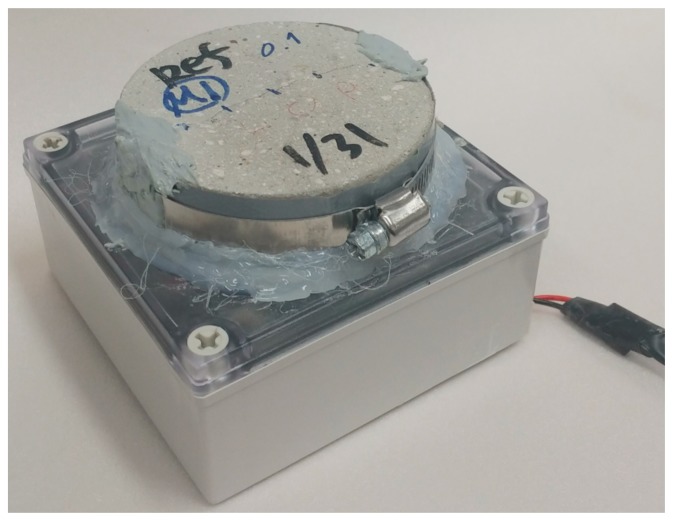
Mortar specimens attached on the airtight container.

**Figure 4 materials-12-03003-f004:**
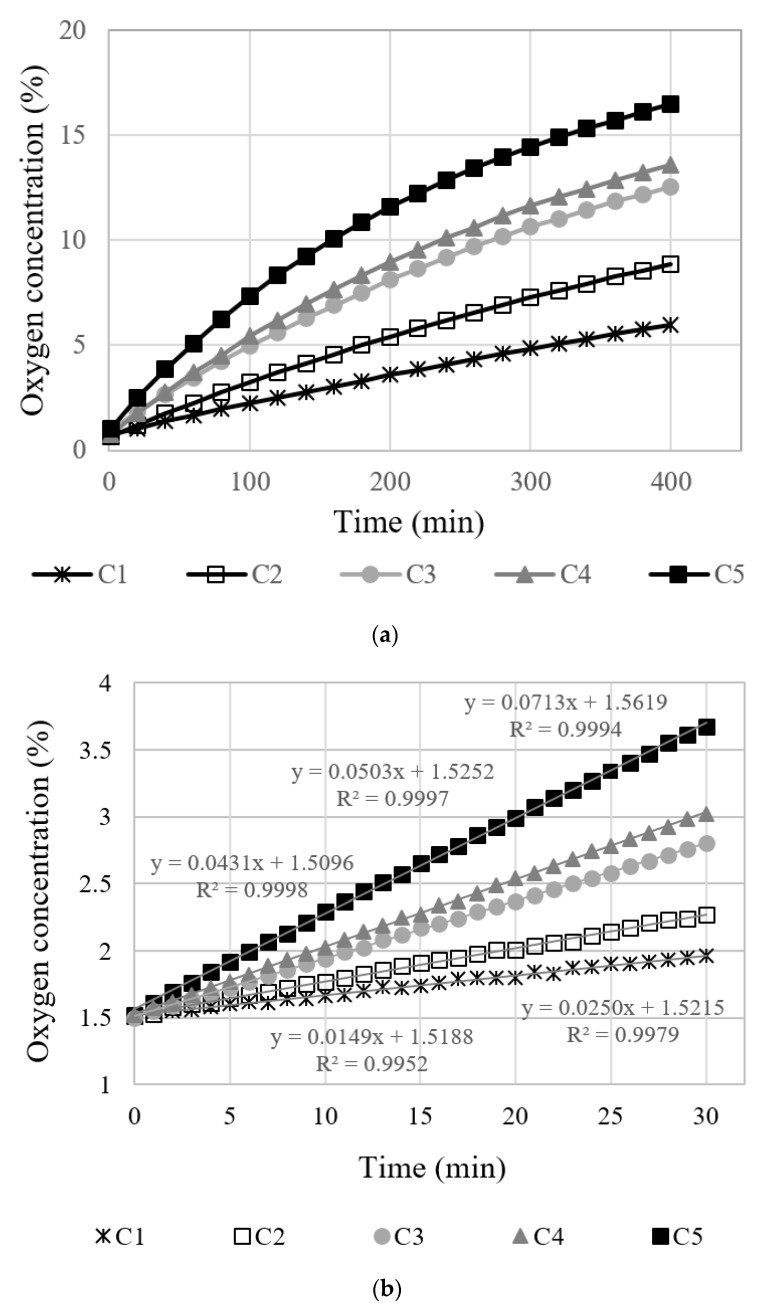
Oxygen concentration of C series specimens with respect to measuring time: (**a**) Overall measurements; (**b**) initial concentration 1.5%, measuring 30 min.

**Figure 5 materials-12-03003-f005:**
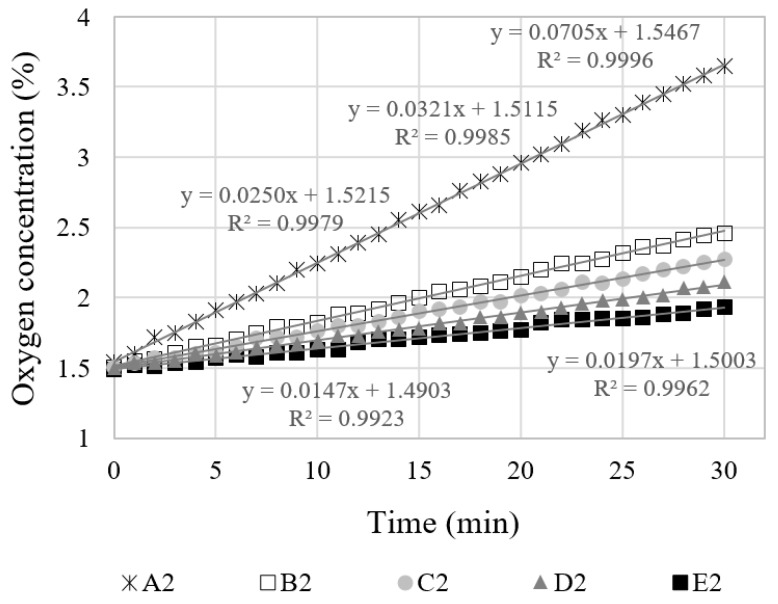
Oxygen concentration with respect to the thickness of specimens, crack width 0.2 mm.

**Figure 6 materials-12-03003-f006:**
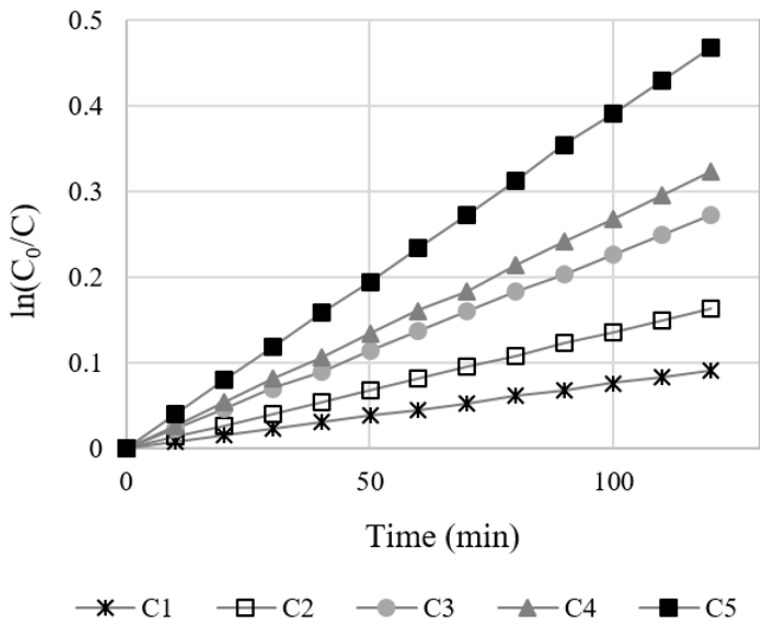
Time versus log of the ratio of concentration relationship.

**Figure 7 materials-12-03003-f007:**
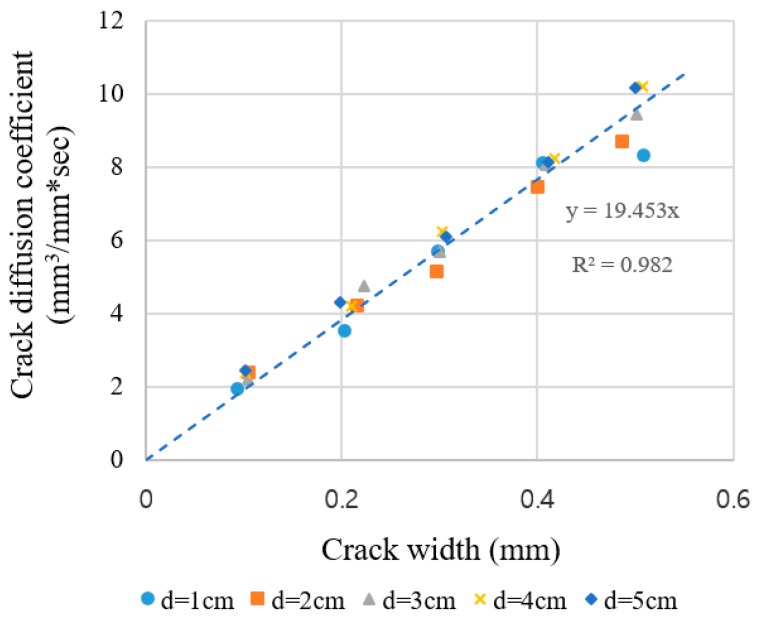
Crack diffusion coefficients versus crack width of the acrylic specimens.

**Figure 8 materials-12-03003-f008:**
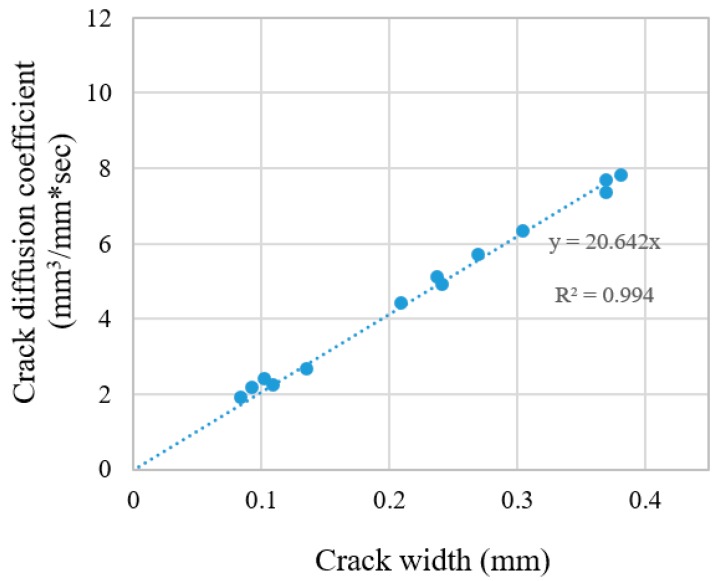
Crack diffusion coefficients versus crack width of the mortar specimens.

**Table 1 materials-12-03003-t001:** Specimen information.

Classification	Depth (mm)	Series Number (Crack Width)				
		1 (0.1 mm)	2 (0.2 mm)	3 (0.3 mm)	4 (0.4 mm)	5 (0.5 mm)
A series	10	0.095	0.203	0.300	0.407	0.510
B series	20	0.106	0.216	0.297	0.401	0.487
C series	30	0.105	0.223	0.301	0.409	0.502
D series	40	0.102	0.210	0.303	0.418	0.508
E series	50	0.101	0.198	0.307	0.411	0.511

**Table 2 materials-12-03003-t002:** Calculated oxygen diffusion coefficient from acrylic specimens.

Classification	Depth (mm)	Series Number (Crack Width)				
		1 (0.1 mm)	2 (0.2 mm)	3 (0.3 mm)	4 (0.4 mm)	5 (0.5 mm)
A series	10	0.201	0.173	0.189	0.199	0.163
B series	20	0.210	0.212	0.189	0.197	0.188
C series	30	0.221	0.211	0.206	0.197	0.201
D series	40	0.222	0.216	0.198	0.198	0.203
E series	50	0.224	0.194	0.182	0.186	0.182

**Table 3 materials-12-03003-t003:** Predicted crack width of acrylic specimens.

Classification		Crack Diffusion Coefficient (mm^3^/mm·s)	Crack Width			
			Measured $w_{m}$ (mm)	Predicted $w_{p}$ (mm)	$\left|w_{m}-w_{p}\right|$ (mm)	$\frac{\left|w_{m}-w_{p}\right|}{w_{p}}$
A Series	A1	1.900	0.098	0.095	0.003	3.24%
	A2	3.512	0.181	0.203	0.022	12.43%
	A3	5.663	0.291	0.300	0.008	2.89%
	A4	8.099	0.416	0.407	0.010	2.37%
	A5	8.305	0.427	0.510	0.083	19.34%
B Series	B1	2.372	0.122	0.106	0.016	13.06%
	B2	4.193	0.216	0.216	0.000	0.21%
	B3	5.120	0.263	0.297	0.034	12.85%
	B4	7.439	0.382	0.401	0.018	4.73%
	B5	8.662	0.445	0.487	0.041	9.26%
C Series	C1	2.195	0.113	0.105	0.008	7.41%
	C2	4.736	0.243	0.223	0.020	8.41%
	C3	5.679	0.292	0.301	0.009	2.94%
	C4	8.037	0.413	0.409	0.004	1.00%
	C5	9.437	0.485	0.502	0.016	3.38%
D Series	D1	2.361	0.121	0.102	0.019	15.96%
	D2	4.209	0.216	0.210	0.007	3.18%
	D3	6.240	0.321	0.303	0.018	5.70%
	D4	8.220	0.423	0.418	0.005	1.19%
	D5	10.181	0.523	0.508	0.016	3.03%
E Series	E1	2.448	0.126	0.101	0.025	19.73%
	E2	4.281	0.220	0.198	0.022	10.04%
	E3	6.093	0.313	0.307	0.006	1.98%
	E4	8.129	0.418	0.411	0.007	1.77%
	E5	10.174	0.523	0.501	0.023	4.30%

**Table 4 materials-12-03003-t004:** Predicted crack width of mortar specimens.

Classification	Crack Width			
	Measured $w_{m}$ (mm)	Predicted $w_{p}$ (mm)	$\left|w_{m}-w_{p}\right|$ (mm)	$\frac{\left|w_{m}-w_{p}\right|}{w_{p}}$
M1	0.084	0.091	0.007	8.15%
M2	0.093	0.105	0.012	11.64%
M3	0.103	0.115	0.012	10.80%
M4	0.110	0.108	0.002	1.86%
M5	0.136	0.129	0.007	5.17%
M6	0.210	0.214	0.004	2.04%
M7	0.238	0.248	0.009	3.80%
M8	0.242	0.237	0.005	1.95%
M9	0.270	0.276	0.006	2.31%
M10	0.305	0.307	0.002	0.54%
M11	0.370	0.371	0.001	0.33%
M12	0.370	0.356	0.014	3.91%
M13	0.382	0.378	0.004	1.02%

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
