# Peer review of "Crack Width Estimation of Mortar Specimen Using Gas Diffusion Experiment"

_materials, 2019, doi:10.3390/ma12183003_

Round 1

Reviewer 1 Report

More related references are needed You may refine or delete the section 2.3 What’s the purpose of regression in Figure 8? More discussion for test results? Not a report. Eliminate the influence of viscosity? Material leaching? Really?

Reviewer 2 Report

Interesting paper on solving a practical problem. Very relevant to scientific community. Some suggestions:

crack width 'w' is not defined in manuscript. The authors should define it (perhaps line 131).  Fabrication procedure for acrylic samples is not adequately clear from the manuscript.  Authors should include photographs of acryclic specimen and concrete specimen with Nitrogen injection.  The authors should comment on effectiveness of seal and include experiments with uncracked acrylic specimen to make sure that the quality of seal is good enough. 

Round 2

Reviewer 1 Report

Authors did not answer all questions and still need to revise the manuscript. More RELATED references here It is better to refine or delete section 2 Still, need to explain the Regression in Figure 8 or delete it? More DISCUSSIONS for research? Not a report! Eliminate the influence of viscosity? Refine the conclusions

Reviewer 2 Report

Changes have been made as requested, so recommend to publish. 

Author Response

Thank you for your kind comment

Reviewer 3 Report

The authors have properly addressed all of my comments in the first round of
review. I recommend publication in the present form. No further review is necessary.

Author Response

Thank you for your kind comment

Round 3

Reviewer 1 Report

Okay